# IFN-λ1 Displays Various Levels of Antiviral Activity In Vitro in a Select Panel of RNA Viruses

**DOI:** 10.3390/v13081602

**Published:** 2021-08-12

**Authors:** Marina Plotnikova, Alexey Lozhkov, Ekaterina Romanovskaya-Romanko, Irina Baranovskaya, Mariia Sergeeva, Konstantin Kаа, Sergey Klotchenko, Andrey Vasin

**Affiliations:** 1Smorodintsev Research Institute of Influenza, Russian Ministry of Health, 197376 St. Petersburg, Russia; biomalinka@mail.ru (M.P.); aswert6@mail.ru (A.L.); romromka@yandex.ru (E.R.-R.); irina.baranovskaja.1992@gmail.com (I.B.); mari.v.sergeeva@gmail.com (M.S.); vasin_av@spbstu.ru (A.V.); 2Institute of Biomedical Systems and Biotechnologies, Peter the Great St. Petersburg Polytechnic University, 195251 St. Petersburg, Russia; 3Chumakov Federal Scientific Center for Research and Development of Immune-and-Biological Products RAS, 108819 Moscow, Russia; kaa_23@mail.ru; 4Scientific and Educational Center for Biophysical Research in The Field of Pharmaceuticals, Saint Petersburg State Chemical Pharmaceutical University, 197022 St. Petersburg, Russia

**Keywords:** IFN-lambda, influenza virus, SARS-CoV-2, CHIKV, antiviral effect

## Abstract

Type III interferons (lambda IFNs) are a quite new, small family of three closely related cytokines with interferon-like activity. Attention to IFN-λ is mainly focused on direct antiviral activity in which, as with IFN-α, viral genome replication is inhibited without the participation of immune system cells. The heterodimeric receptor for lambda interferons is exposed mainly on epithelial cells, which limits its possible action on other cells, thus reducing the likelihood of developing undesirable side effects compared to type I IFN. In this study, we examined the antiviral potential of exogenous human IFN-λ1 in cellular models of viral infection. To study the protective effects of IFN-λ1, three administration schemes were used: ‘preventive’ (pretreatment); ‘preventive/therapeutic’ (pre/post); and ‘therapeutic’ (post). Three IFN-λ1 concentrations (from 10 to 500 ng/mL) were used. We have shown that human IFN-λ1 restricts SARS-CoV-2 replication in Vero cells with all three treatment schemes. In addition, we have shown a decrease in the viral loads of CHIKV and IVA with the ‘preventive’ and ‘preventive/therapeutic’ regimes. No significant antiviral effect of IFN-λ1 against AdV was detected. Our study highlights the potential for using IFN-λ as a broad-spectrum therapeutic agent against respiratory RNA viruses.

## 1. Introduction

Interferons (IFNs) play a critical role in the immune response, by suppressing the spread of viral infection in the early stages of illness, and they form the first line of defense in mammals against viral infection [1,2]. Type III IFN (IFN-λ) is a group of IFNs related to IFN-α/β, showing similar antiviral effects [3,4]. Four subtypes of IFN-λ have been found in humans: IFN-λ1 (IL-29); IFN-λ2 (IL-28A); IFN-λ3 (IL-28B); and IFN-λ4 [5]. The action of IFN-λ is realized through the heterodimeric IFNLR receptor, which consists of the IFNλR1 and IL10R2 subunits. The IL10R2 subunit is also part of receptor complexes for IL-10, IL-22, and IL-26; it is expressed in the cells of various tissues [6]. In contrast, the IFNλR1 subunit exhibits limited cellular distribution. A high level of IFNλR1 has been noted in the lungs, intestines, liver, and upper epidermis [7].

IFNλR1 expression is mainly limited to epithelial cells [8], keratinocytes [9], differentiated dendritic cells (pDC and cDC) [10,11], and hepatocytes [12]. Consequently, mucous membranes of the respiratory and gastrointestinal tracts are tissues that are mainly targeted by IFN-λ [8]. This tissue specificity correlates with the antiviral activity of IFN-λ, which manifests itself mainly in relation to viruses with a high tropism for epithelial tissues [7].

It is known that IFN-λ, like IFN-α/β, induces JAK/STAT-mediated activation of interferon-stimulated genes (ISGs), but the dynamics of this activation are different [5,13]. The transcriptional response to IFN-λ is generally weaker than to IFN-α/β, but is characterized by a longer duration [14,15]. The use of IFN-λ as a potential antiviral drug has several advantages over IFN-α/β. Using in vivo models, it has been shown that the use of IFN-α/β for influenza (viral infection) treatment has led to conflicting results [16]. Side effects were explained by the immunomodulatory effects of IFN-α: recruitment of cells of the innate immune system to the site of infection (pDC, monocytes); increased levels of proinflammatory cytokines and chemokines in bronchoalveolar lavage (IL-6, IP-10, MCP-1, MIP-1α); and increased respiratory epithelial cell apoptosis frequency [17].

Transcriptome analysis has shown that there is a cluster of ISGs, the expression of which is specifically induced by IFN-α, but not by IFN-λ. This cluster is associated with: recruitment of immune cells; hypercytokinemia; and hyperchemokinemia [17,18]. These make significant contributions to the development of cytokine storm and inflammatory processes in lung tissues [19,20]. Therefore, although IFN-α/β are potent immunomodulators in themselves [21], the addition of exogenous IFN-α/β can lead to excessive activation of inflammatory processes [17,18]. In contrast, IFN-λ can be viewed as a potentially promising therapeutic agent.

The most urgent question regards the possible use of IFN-λ as a therapeutic agent against SARS-CoV-2. Interestingly, increased serum IFN-λ levels were found only in non-intensive care unit (non-ICU) COVID-19 patients, while more severe ICU patients did not show increased serum IFN-λ [22]. Consequently, a severe course of infection is associated with impaired functioning of the innate immune system and activation of ISGs. The administration of exogenous IFN-λ appears to be a promising approach to alleviate the disease course. Clinical studies on the effect of PEG-IFN-λ1, on the recovery dynamics of patients with SARS-CoV-2 infection, have been carried out. There was a slight decrease in viral load on the seventh day after the start of the experiment, but there were no differences in clinical symptoms between the PEG-IFN-λ1 and placebo groups [23]. For a more accurate assessment of the therapeutic potential of IFN-λ1, we carried out additional in vitro studies.

## 2. Materials and Methods

### 2.1. Cell Lines

The A549 (ATCC CCL-185) and Vero (ATCC CCL-81) cell lines were obtained from the American Type Culture Collection (ATCC, Manassas, VA, USA). A549 cells (human type II alveolar epithelial line) were maintained by continuous culture in DMEM/F12K (Gibco, Grand Island, NE, USA) supplemented with GlutaMAX (to 1×) and 10% fetal bovine serum (Biowest, Riverside, CA, USA; South America). Vero cells (African green monkey kidney epithelial line) were maintained in AlphaMEM (Biolot, St. Petersburg, Russia) or Eagle’s MEM (for CHIKV infection) containing 10% fetal bovine serum (Biowest, Riverside, CA, USA; South America). Cells were cultured at 37 °C (5% CO_2_ with humidification).

### 2.2. Viruses

The influenza virus strains A/California/07/09 (H1N1pdm09) and A/PR/8/34 (H1N1) (IVA) were obtained from the Virus and Cell Culture Collection of the Smorodintsev Research Institute of Influenza (St. Petersburg, Russia). Influenza viruses were grown in 11-day-old embryonated eggs, purified by sucrose gradient, and stored at −80 °C. The infectivity of the viruses stocks in MDCK cells was 3.2 × 10^7^ TCID_50_/mL for H1N1pdm09 strain and 3.2 × 10^8^ TCID_50_/mL for H1N1 strain.

A chikungunya virus (CHIKV) strain was derived and characterized by the Chumakov Federal Scientific Center for Research and Development of Immune-and-Biological Products, Russian Academy of Sciences (Moscow, Russia). The infectivity of the CHIKV virus stock in Vero cells was 2.95 × 10^8^ pfu/mL or 1.10 × 10^8^ TCID_50_/mL.

SARS-CoV-2 virus (hCoV-19/Russia/SPE-RII-3524V/2020 (GISAID ID EPI_ISL_415710)) was isolated from a patient oropharyngeal swab in Vero cells (ATCC CCL-81) in Smorodintsev Research Institute of Influenza (St. Petersburg, Russia). Cell culture was inoculated for 2 h with swab material diluted 1:10 in AlphaMEM (Biolot) supplemented with 2% HI-FBS (Gibco), 1% anti-anti (Gibco). Incubation then proceeded for 3–4 days, until the appearance of cytopathic effects. Two more passages were made in Vero cells to generate the virus working stock (V3). Virus was titrated in Vero cells by the standard limiting-dilution method in 96-well plates. The plate readout was performed microscopically four days post inoculation, and productive infection was recorded in case of visible CPE. The infectivity of the SARS-CoV-2 virus stock in Vero cells was 1.26 × 10^7^ TCID_50_/mL. Infections with SARS-CoV-2 virus were performed in a BSL-3 facility.

Adenovirus type 5 (strain Adenoid 75) was obtained from the Virus and Cell Culture Collection of the Smorodintsev Research Institute of Influenza (St. Petersburg, Russia). AdV working stock was generated by infecting A549 cells at a multiplicity of infection (MOI) of 0.001 for 72 h. Supernatant was then clarified by centrifugation, aliquoted, and stored at −80 °C. The infectivity of the viruses stocks in A549 cells was 3.2 × 10^6^ TCID_50_/mL.

### 2.3. Influenza Virus Infection Assays

For A549 experiments, 1.8 × 10^4^ cells/well were seeded overnight into 96-well plates before treatment with the indicated IFN-λ. For ‘preventive’ and ‘preventive/therapeutic’ treatment, cells were then stimulated with IFN-λ1 at concentrations of 10, 100 or 500 ng/mL. For ‘therapeutic’ treatment, the medium was replaced with DMEM/F12K (Gibco) without serum. After 24 h of stimulation, cells were infected with A/California/07/09 (H1N1pdm09) at 10^–1^–10^–3^ dilution. After 1 h of virus adsorption, inoculum was removed, and cells were allowed to grow in medium containing IFN-λ1 at concentrations of 10, 100 or 500 ng/mL (for ‘therapeutic’ and ‘preventive/therapeutic’ scheme) or without IFN-λ1 (for the ‘preventive’ scheme). Viral antigen was measured at 24 hpi by In-Cell ELISA with anti-NP antibodies.

Additionally, RT-qPCR was used to study replication of the IVA genome (M-protein gene) in A549 cells in the period from 3–24 h in response to treatment with IFN-λ1 (500 ng/mL) in the ‘preventive/therapeutic’ scheme and without it.

For Vero infection, 2 × 10^4^ cells/well were seeded overnight into 96-well plates before treatment with the indicated IFN-λ1. For ‘preventive’ and ‘preventive/therapeutic’ treatment, cells also were stimulated IFN-λ1 at concentrations of 10, 100 or 500 ng/mL. For ‘therapeutic’ treatment, the medium was replaced with AlphaMEM (Biolot) without serum. After 24 h of stimulation, cells were infected with A/PR/8/34 (H1N1) diluted 10-fold from neat to 10^–5^. After 1 h of virus adsorption, inoculum was removed, and cells were allowed to grow in medium containing IFN-λ1 at concentrations of 10, 100 or 500 ng/mL (for ‘therapeutic’ and ‘preventive/therapeutic’ scheme) or without IFN-λ1 (for ‘preventive’ scheme). For multicycle IVA replication, the medium after infection additionally contained TPCK-treated trypsin (Thermo Fisher Scientific, Waltham, MA, USA) at a final concentration of 1 μg/mL. Viral infectious activity, with or without IFN-λ1, was measured by endpoint dilution assay at 72 hpi. Presence of virus was assessed visually (observable cytopathogenic effects) or by In-Cell ELISA with anti-NP antibodies (for single-cycle IVA replication). The 50% tissue culture infectious dose (TCID_50_) was calculated by the Reed and Muench method [24,25].

### 2.4. SARS-CoV-2 Infection Assay

Vero cells were grown to 90 to 100% confluence in 96-well plates. For ‘preventive’ and ‘preventive/therapeutic’ regimes, cells were stimulated with IFN-λ1 at concentrations of 10, 100 or 500 ng/mL. For ‘therapeutic’ treatment, the medium was replaced with MEM (Biolot) without serum. After 24 h of stimulation, cells were infected with SARS-CoV-2 at 10^–2^–10^–5^ dilution. After 1 h of virus adsorption, inoculum was removed, and cells were allowed to grow in medium containing IFN-λ1 at concentrations of 10, 100 or 500 ng/mL (for ‘therapeutic’ and ‘preventive/therapeutic’ scheme) or without IFN-λ1 (for the ‘preventive’ scheme). Viral infectious activity, with or without IFN-λ1, was measured by endpoint dilution assay at 72 hpi. Presence of virus was assessed visually (observable cytopathogenic effects). The 50% tissue culture infectious dose (TCID_50_) was calculated by the Reed and Muench method [24,25].

### 2.5. CHIKV Infection Assay

Vero cells were grown to 90 to 100% confluence in 24-well plates. For ‘preventive’ and ‘preventive/therapeutic’ treatments, cells were stimulated IFN-λ1 at concentrations of 10, 100 or 500 ng/mL. For ‘therapeutic’ treatment, the medium was replaced with Eagle’s MEM (Biolot) without serum. After 24 h of stimulation, cells were infected with CHIKV at 25 pfu per well. After 1 h of virus adsorption, inoculum was removed, and overlay medium containing IFN-λ1 at concentrations of 10, 100 or 500 ng/mL (for ‘therapeutic’ and ‘preventive/therapeutic’ scheme) or without IFN-λ1 (for virus control and ‘preventive’ scheme) was added to wells. Overlay medium contained 1% methylcellulose (Sigma-Aldrich) in Eagle’s MEM (Biolot) supplemented with 2% FBS. After 4 days of incubation, cells were fixed and stained with 2% crystal violet in 20% ethanol. Plaques were counted to calculate the inhibitory concentration of the IFN-λ1 against CHIKV.

### 2.6. Adenovirus Infection

A549 cells were seeded overnight into 96-well plates before treatment with the indicated IFN-λ1. For ‘preventive’ and ‘preventive/therapeutic’ treatments, cells were then stimulated with IFN-λ1 at concentrations of 10, 100 or 500 ng/mL. For ‘therapeutic’ treatment, the medium was replaced with DMEM/F12K (Gibco) without serum. After 24 h of stimulation, cells were infected with AdV at 10^–3^–10^–5^ dilution. After 1 h of virus adsorption, inoculum was removed, and cells were allowed to grow in medium containing IFN-λ1 at concentrations of 10, 100 or 500 ng/mL (‘preventive’ and ‘preventive/therapeutic’ schemes) or without IFN-λ1 (for preventive scheme). Viral load was measured at 72 hpi by In-Cell ELISA with the anti-hexon antibodies. The same experiments were also performed in 24-well plates. Viral antigen was measured at 24 hpi by In-Cell ELISA with anti-hexon antibodies. Additionally, RT-qPCR was used to study replication of the AdV genome (hexon gene) in A549 cells in the period from 3–24 h in response to treatment with IFN-λ1 (500 ng/mL) in the ‘preventive/therapeutic’ scheme and without it.

### 2.7. In-Cell ELISA

At selected time points, inoculum was removed from cell plates, and cells in monolayer were fixed by cold 80% acetone in DPBS for 30 min in +4 °C. After washing with a 1× PBST solution (0.05% Tween 20), wells were blocked with 5% blotting-grade blocker (Bio-Rad, Hercules, CA, USA), diluted in 1× PBST, for 1 h at room temperature. After plate washing, 100 μL of (1 μg/mL) mAb specific to viral NP (for IVA detection) or hexon (for AdV detection) were added to the wells, and the plate reincubated at room temperature for 2 h. Binding was detected using GAM-HRP (Bio-Rad) secondary antibodies diluted 1:500 in 1× PBST (30 min incubation at room temperature). Peroxidase reaction was performed using the TMB Peroxidase EIA Substrate Kit (Bio-Rad). After 10 min, reactions were stopped by addition of 50 μL of 2 N H_2_SO_4_ to each well. Optical densities were measured at 450 nm (OD_450_) on a CLARIOstar plate photometer (BMG LABTECH, Ortenberg, Germany).

### 2.8. PCR Analysis

Total RNA was isolated using the Trizol Reagent (Invitrogen, Austin, TX, USA) with full adherence to the manufacturer’s recommendations. Two micrograms of total RNA were treated by DNase (Promega, Madison, WI, USA) and then directly reverse-transcribed using oligo-dT_16_ primers and MMLV reverse transcriptase (Promega). Complementary DNA synthesis was carried out at 42 °C for 60 min; products were stored at −20 °C until use. qPCR was performed using the 2× BioMaster HS-qPCR reagent (BioLabMix, Novosibirsk, Russia) and previously-developed primers (Appendix A). Relative expression values were calculated by the ΔΔCt method using GAPDH as a normalization gene.

### 2.9. Statistical Analysis

Statistical analyses were performed using GraphPad Prism 6 (GraphPad 6.0 Software). Data were log transformed and analyzed by either one-way or two-way analysis of variance (ANOVA) for multiple comparisons. *p* values for significance are given in the figure legends.

## 3. Results

### 3.1. Antiviral Effect against Influenza

To study the potential antiviral effects of type III interferon, we used recombinant (human) IFN-lambda protein previously obtained and characterized by the authors [26]. The amino acid sequence of recombinant 6× His-hIFN-λ1 protein was confirmed using MALDI-TOF mass spectrometry. The therapeutic effect of IFN-λ1 against influenza A (IVA) viruses was investigated in the permissive cell cultures A549 and Vero (Appendix A). The toxicity of IFN-λ1 in these cell lines was examined 24 h after stimulation. Cell viability measurement was carried out using the (tetrazolium dye) MTT assay (3-(4,5-dimethylthiazol-2-yl)-2,5-diphenyltetrazolium bromide). Our results indicate an absence of significant IFN-λ1 toxicity at a wide range of dilutions. The IC_50_ of IFN-λ1 for A549 cells exceeded 25 μg/mL. For Vero, it exceeded 2 μg/mL (Appendix A).

Study of antiviral activity in A549 cells was carried out using three schemes of IFN-λ1 administration at concentrations of 10 ng/mL, 100 ng/mL, and 500 ng/mL. Viral load was assessed based on the amount of viral nucleoprotein in supernatants obtained from cells 24 h after infection. As seen in Figure 1a, a virus-inhibiting effect was observed only with IFN-λ1 administration 24 h before infection, both in the ‘preventive’ and ‘preventive/therapeutic’ regimes. The ‘therapeutic’ IFN-λ1 treatment did not lead to a significant decrease in the level of viral protein in the supernatants. In a number of experiments in A549 cells (Appendix A), we showed a tendency towards an increase in the production of viral NP protein in response to IFN-λ1 treatment of cells immediately after infection. This effect was reproduced in several experiments (not shown) when IFN-λ1 concentration was increased to 2500 ng/mL.

When evaluating the replication of IVA viral RNA in IFN-λ-stimulated A549 cells in the ‘preventive/therapeutic’ regime, it was shown (Appendix A) that, already three hours after infection, IFN-λ1 treatment leads to a decrease in the number of copies of viral RNA by about fivefold compared to the control. In general, the trend of a decrease in RNA replication (five- to ninefold) was maintained throughout the entire time period considered.

In experiments in Vero cells, the antiviral effect of IFN-λ1 was studied with single-cycle (Appendix A) and multiple-cycle IVA replication. Multiple-cycle IVA replication (Figure 1b) was achieved by adding trypsin to the medium after infection at a concentration of 1 μg/mL. Under these conditions (multiple-cycle IVA), the calculated viral load was approximately 2 Log (TCID_50_) higher, compared to single-cycle. The antiviral effect of IFN-λ1 was only observed with the ‘preventive’ and ‘preventive/therapeutic’ administration regimens with single cycle IVA replication. With multiple-cycle IVA replication in cells, despite a trend towards a decrease in viral load upon IFN-λ1 treatment, the changes were statistically significant only for IFN-λ1 at 500 ng/mL in all treatment schemes.

### 3.2. Changes in ISG Expression after IFN-Λ1 Stimulation

The expression of selected interferon-stimulated genes (ISGs) was assessed to confirm: the specificity of any effects seen; and to confirm that they are due to the action of IFN-λ1. As shown in Figure 2, addition of recombinant IFN-λ1 led to an increase in the expression of the canonical antiviral ISGs MxA, PKR, and OAS-1. Meanwhile, the expression level of MxA mRNA, which has a direct antiviral effect against influenza [27], increased more than 500-fold (relative to nonstimulated cells) and more than 10-fold (relative to cells infected with IVA). It is noteworthy that the expression of PKR and OAS-1 in the case of IVA infection was suppressed. Similarly, the expression of SOCS-1 was determined, as it carries out negative regulation of proinflammatory cytokines secreted by cells in response to IFN-λ1 stimulation. We have shown an increase in SOCS-1 expression in response to both IVA infection and IFN-λ1 stimulation of cells. In addition, the expression of Rig-1, a cellular sensor involved in the innate immune response to influenza, was also found to be altered in response to IFN-λ1 stimulation. Interestingly, the expression profiles of the considered genes upon ‘preventive’ administration of IFN-λ1 followed by infection with IVA’ generally differed little from cells that were not infected with IVA. The administration of IFN-λ1 generally led to more powerful expression of the considered genes, including MxA, Rig-1, and SOCS-1, which, in general, increased upon IVA infection.

Since the Vero cell lineage, which has *Chlorocebus sabaeus* origin (not *Homo sapiens*), was also used to assess the antiviral activity, we evaluated the ability of the human recombinant IFN-λ1 protein to induce ISGs in these cells. In Vero cells, IFN-λ1 also induced the expression of MxA, PKR, OAS-1, as well as SOCS-1 and Rig-1. The most significant changes were in MxA expression. Upon treatment with IFN-λ1, the average mRNA level of this gene in Vero cells increased more than 4000-fold (Appendix A).

### 3.3. Antiviral Action against SARS-CoV-2

The antiviral effect of IFN-λ1 against SARS-CoV-2 was investigated in the Vero cell line. Recombinant IFN-λ1 was administered at three concentrations using three dosing regimens. As seen in Figure 3, IFN-λ1 at the maximum studied dose led to a statistically significant decrease in the SARS-CoV-2 viral load in Vero cells in all the considered administration regimens. It is curious that, at the minimum dose used (10 ng/mL), IFN-λ1 did not have an antiviral effect. However, IFN-λ1 stimulation of cells, at both 100 ng/mL and 500 ng/mL, resulted in a similar decrease in viral titer: about 1.5 logs for ‘preventive’ treatment; and 3.5 logs for ‘preventive/therapeutic’ treatment. Despite the statistically significant decrease in viral load with the therapeutic IFN-λ1 regime (at 500 ng/mL) shown, the inhibitory effect on viral reproduction was rather weak.

### 3.4. Antiviral Action against Chikungunya Virus

We also investigated the antiviral effect of recombinant IFN-λ1 against the RNA-containing (fourth Baltimore group) arbovirus CHIKV (Figure 4). The data obtained showed that the introduction of IFN-λ1 to cells infected with CHIKV, according to the treatment scheme, led to a weak, but statistically significant, increase in the CHIKV viral load in the cells. This effect was observed with all IFN-λ1 doses studied. In addition, the use of IFN-λ1 according to the ‘preventive’ and ‘preventive/therapeutic’ schemes led to a decrease in viral load by about 20 pfu. It is interesting to note that, as with SARS-CoV-2, IFN-λ1 at 10 ng/mL did not have a significant inhibitory effect against CHIKV when administered as ‘preventive’, but led to an increase in the viral load when administered according to the ‘therapeutic’ regimen.

### 3.5. Antiviral Action against Adenovirus

Finally, we evaluated the antiviral potential of recombinant human IFN-λ1 against DNA-containing, nonenveloped adenovirus (AdV). Unfortunately, administration of IFN-λ1 (at 10, 100, and 500 ng/mL), in accordance with all three treatment regimens (‘preventive’, ‘preventive/therapeutic’, ‘therapeutic’), did not lead to a significant decrease in AdV production in A549 cells (Appendix A). Viral load was measured by visual determination of TCID_50_, as well as by ELISA (of AdV hexon), at 24 hpi and 72 hpi.

To clarify and confirm the absence of an antiviral effect of IFN-λ1 against AdV, a curve was constructed reflecting the kinetics of the accumulation of viral RNA (hexon gene) over time. As shown in Figure 5b (Appendix A), the treatment of A549 cells with IFN-λ1 did not lead to a decrease in the replication of the AdV viral genome.

## 4. Discussion

The presented study focuses on features that relate to the use of human IFN-λ1 as a potential antiviral agent against specific respiratory viruses. We have previously obtained recombinant human interferon-λ1 [26]. It is known that both IFN-α/β and IFN-λ induce the expression of similar groups of ISGs (interferon stimulated genes) [21,28], which include such canonical antiviral genes as MxA, OAS, PKR, IFITM, and ISG15 [29,30]. In our work, we showed the ability of IFN-λ to activate the expression of ISGs such as MxA, PKR, OAS-1, and Rig-1 in A549 cells. It is interesting to note that PKR and OAS-1 expression was suppressed when cells were IVA-infected and did not differ significantly from control cells. Stimulation of cells with IFN-λ1 with subsequent IVA infection also led to an increase in the expression of ISGs, which disables the ability of IVA to suppress the innate immune response.

It is known that the IVA NS1 protein is an agonist of the cellular IFN system. The NS1 protein binds to viral dsRNAs, rendering them invisible in the infected cell, and inhibits PKR protein kinase activation [31,32], thus preventing protein translation inhibition (in the infected cell) and delaying apoptosis. In addition, the NS1 protein also inhibits the ability of the Rig-1 intracellular sensor to detect viral RNAs and trigger the IFN pathway. According to our results, prophylactic administration of IFN-λ1 creates a specific antiviral state in the cell which, upon further infection, practically does not lend itself to immunomodulation by IVA.

We have shown an increase in the expression of SOCS-1 mRNA, which takes part in a negative feedback loop to attenuate cytokine signaling, including as a mechanism by which influenza viruses inhibit the host antiviral response. It has been shown in the literature that IFN-λ induces early expression of SOCS-1, a negative regulator of JAK/STAT signaling. The peak of inhibitor expression occurs 4–8 h after IFN-λ exposure, while the SOCS-1 mRNA level increases starting from 16 h after IFN-α administration [13]. In this study, only SOCS-1 showed a cumulative effect, and its expression upon ‘IVA infection of IFN-λ stimulated cells’ was threefold higher compared to ‘stimulated, uninfected’ cells.

Numerous studies have shown that IFN-λ, being a nonspecific activator of a large group of ISGs, exhibits antiviral activity against a whole group of negative strand ssRNA viruses: IVA [18,21,33,34,35,36,37,38]; IBV [39]; RSV [2,40,41]; and LCMV [42]. Some positive strand ssRNA viruses are also affected, such as: HCV [14,43,44,45]; Dengue virus [46,47]; and rhinovirus [48,49]. Here, we showed that IFN-λ suppressed the replication of IVA, CHIKV, and SARS-CoV-2 viruses during the ‘preventive’ and ‘preventive/therapeutic’ regimes.

Currently, there is evidence of the antiviral activity of IFN-λ, both in vitro and in vivo, in relation to SARS-CoV-2. Others have shown that type I and type III IFNs restrict SARS-CoV-2 replication in primary human bronchial epithelial cells [50]. In our experiments in Vero cells, we observed a decrease in SARS-CoV-2 viral load when using three IFN-λ administration regimens (‘preventive’, ‘preventive/therapeutic’, ‘therapeutic’). The Vero cell line used is defective in type I interferons such as α-8, α-2, α-1/13, α-6, α-14, α-4, α-17, α-21, ω-1, and β-1 [51]. However, we have shown that stimulation of Vero cells with human recombinant IFN-λ1 (homologous to simian IFN-λ1) is capable of inducing endogenous cellular expression of MxA, PKR, OAS-1, SOCS-1, and Rig-1.

Current literature has shown that the DEG (differently expressed gene) pattern of peripheral white blood cells, isolated from whole blood of SARS-CoV-2 patients, depends on infection severity. Patients with moderate COVID-19 severity had increased levels of ISGs. In severe patients, on the contrary, an extremely weak activation of antiviral ISGs (MxA, IFITM, IFIT-2) was noted, and the production of IFN-α2 at the protein level was reduced (compared with moderate cases). Moreover, all groups infected with SARS-CoV-2 were characterized by undetectably low IFN-β, both at the protein and mRNA levels [52]. Thus, the antiviral response we observed against SARS-CoV-2 may be largely mediated by ISGs induced by type III IFN. Interestingly, the decrease in SARS-CoV-2 viral load did not strongly depend on IFN-λ1 concentration (100 or 500 ng/mL). However, a decrease in viral titer was observed only at IFN-λ1 concentrations exceeding the previously calculated EC_50_ (approximately 10 ng/mL).

We noted an interesting effect when evaluating the antiviral effect of IFN-λ1 against the IVA and CHIKV viruses. IFN-λ1 stimulation of cells according to the ‘therapeutic’ scheme either did not have a therapeutic effect (IVA, Vero cells) or led to a weak increase in the production of IVA (A549 cells) or CHIKV (Vero cells) after 24 hpi. Meanwhile, such administration in the ‘preventive’ or ‘preventive/therapeutic’ schemes limited viral growth. It should be noted that in the proposed scheme, we used single-cycle viral replication to analyze the early cellular response to infection in vivo. Contemporary literature describes cases when treatment of influenza viral infection with IFN-α resulted in enhanced morbidity or mortality [17,53].

It is known that, among the hundreds of proteins induced by interferon, an anti-influenza effect was shown for a subgroup including the Mx gene products, PKR, and the families of IFITM, IFIT and RNase L proteins. With IFN-λ administration according to the ‘therapeutic’ scheme, ISGs likely do not have time to accumulate in the cell in significant amounts to ensure an antiviral effect. At the same time, early viral factors that antagonize the IFN system, such as IVA NS1 protein, successfully inhibit the IFN-λ-induced immune response of the cell. Interestingly, with SARS-CoV-2, treatment with IFN-λ resulted in a mild reduction in viral load.

It is possible that a therapeutic effect of IFN-λ also depends on the rate of accumulation of viral proteins and the timing of viral replication. In our experiments in Vero cell culture, when comparing single-cycle and multiple-cycle IVA replication, we did not find a decrease in viral load after the ‘therapeutic’ administration of IFN-λ. It is also worth noting that, when simulating multiple-cycle IVA replication, cell media after infection contained trypsin at a final concentration of 1 μg/mL. With this content of trypsin, loss of IFN-λ functionality and its degradation can be observed. It should be noted that, even with single-cycle IVA replication in Vero cells, the decrease in viral load was insignificant (no more than 1 Log (TCID_50_) relative to the control). Interestingly, IFN-λ stimulation of Vero cells resulted in a generally less critical decrease in cell viral load. This can probably be explained by the fact that Vero cells are defective in type I IFN.

Interestingly, IFN-λ1 treatment leads to a five- to ninefold decrease in viral RNA replication at least during the production of the first generation of virions. Not much is known in the current literature about the antiviral effect of IFN-λ against DNA viruses, although antiviral activity has been shown against CMV [5], HSV-1 and HSV-2 [54,55], and HBV [56]. In our experiments, stimulation of cells with IFN-λ1 did not lead to any significant decreases in the production of viral antigen (hexon) or viral RNA replication, nor decline in infectious activity. Thus, we found no evidence of restriction of AdV (serotype 5) production in cells treated with IFN-λ1. The lack of an antiviral effect against AdV with exogenous IFN-λ1 may be due to various viral repressors (such as E1A and E1B proteins) of cellular protective mechanisms and ISGs [57,58].

## 5. Conclusions

The work presented was devoted to the study of the antiviral effects of human IFN-λ1 against certain respiratory viruses (IVA, SARS-CoV-2, AdV), as well as the CHIKV arbovirus. According to our results, human IFN-λ1 has the potential to limit the replication of RNA viruses in vitro. However, the data presented here show that treatment of cells by IFN-λ1 immediately after viral invasion can sometimes provoke stimulated viral production, as with CHIKV and less often with IVA. It is noteworthy that such an effect was observed in vitro in a single cell population and is apparently due to the interaction between immunosuppressing viral proteins and effector mechanisms of innate immunity. The molecular basis of this pathogenic potential in the formation of IFN-λ-mediated antiviral responses should be more carefully evaluated using animal models.

## Figures and Tables

**Figure 1 viruses-13-01602-f001:**
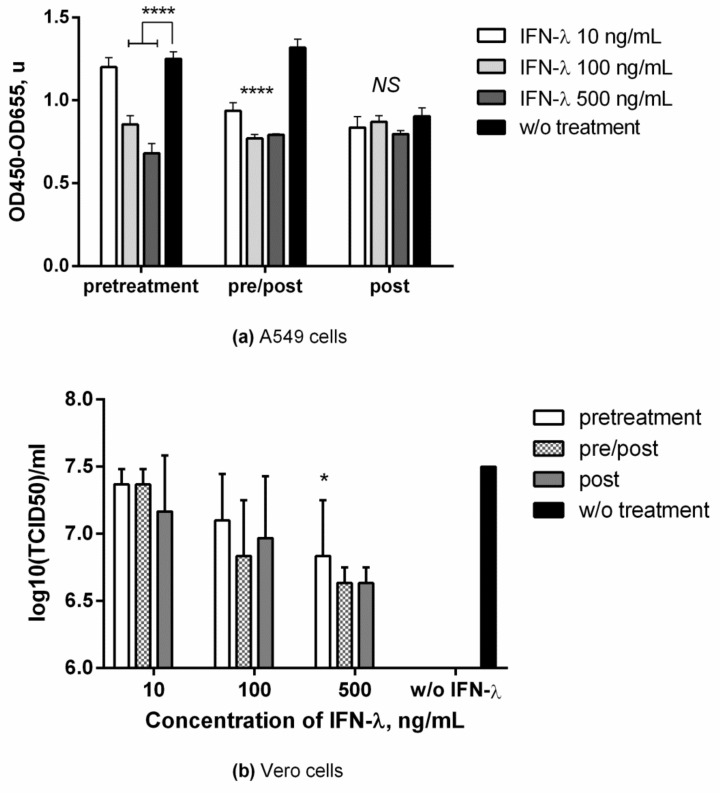
**Inhibition of influenza A replication in different IFN-λ1 treatment regimes**. (**a**) A549 cells were infected by A/California/07/09 (H1N1pdm09) without trypsin. Viral antigen was measured at 24 hpi by In-Cell ELISA with anti-NP antibodies; (**b**) Vero cells were infected by A/PR/8/34 (H1N1) with subsequent addition of TPCK-treated trypsin. Viral titers were determined at 72 hpi. *p*-value significance levels were obtained by comparing groups using the ordinary one-way ANOVA analysis (**a**) or Kruskal-Wallis test (**b**), followed by a pairwise Dunnett’s multiple comparisons test: ****―Adjusted *p* value < 0.0001; *―< 0.05; NS—no significant difference. Data are represented as mean ± SD.

**Figure 2 viruses-13-01602-f002:**
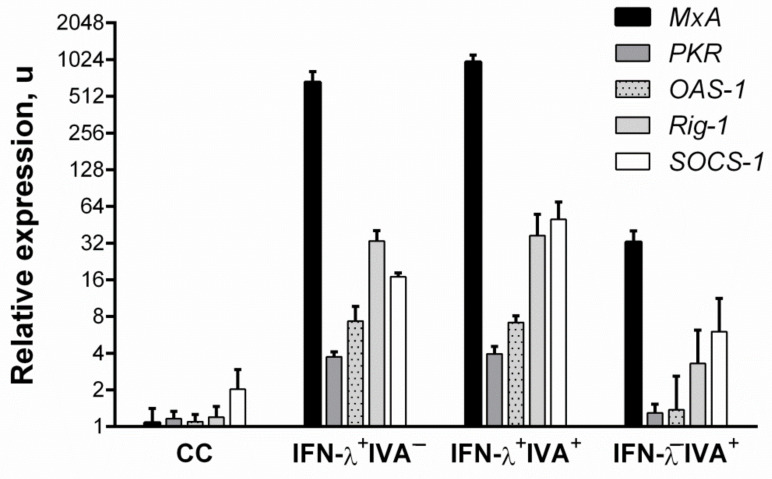
**Interferon-stimulated gene mRNA patterns, in response to IFN-λ1 and IVA stimulation, 24 h after treatment**. A549 cells were used. CC—cell control; IFN-λ^+^IVA^–^—uninfected cells stimulated with IFN-λ1 at 500 ng/mL; IFN-λ^+^IVA^+^—cells infected with A/California/07/09 (H1N1pdm09) at 1 MOI and then stimulated with IFN-λ1 at 500 ng/mL; IFN-λ^−^IVA^+^—cells infected with A/California/07/09 (H1N1pdm09) at 1 MOI. Values are marked as mean ± SD.

**Figure 3 viruses-13-01602-f003:**
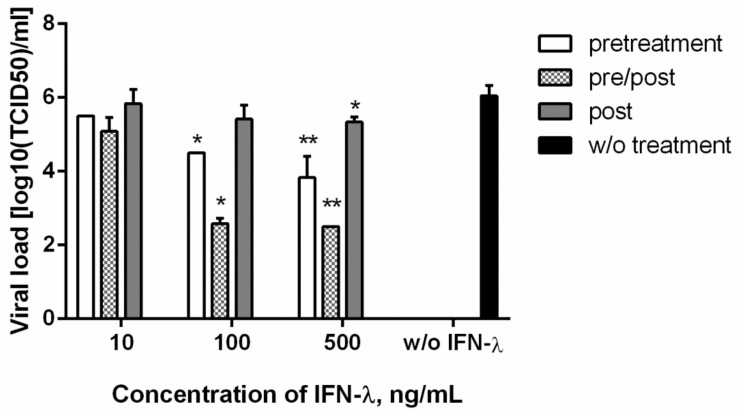
**Inhibition of SARS-CoV-2 replication in different IFN-λ1 treatment regimes**. Vero cells were used. Viral titers were determined at 72 hpi. Statistical significance was determined for cells treated with each IFN-λ1 concentration, compared to untreated cells, by Kruskal-Wallis test (with pairwise Dunnett’s multiple comparisons test) following log transformation: *―Adjusted *p* value < 0.05; **―<0.01; no asterisk means no significant differences. Data are represented as mean ± SD.

**Figure 4 viruses-13-01602-f004:**
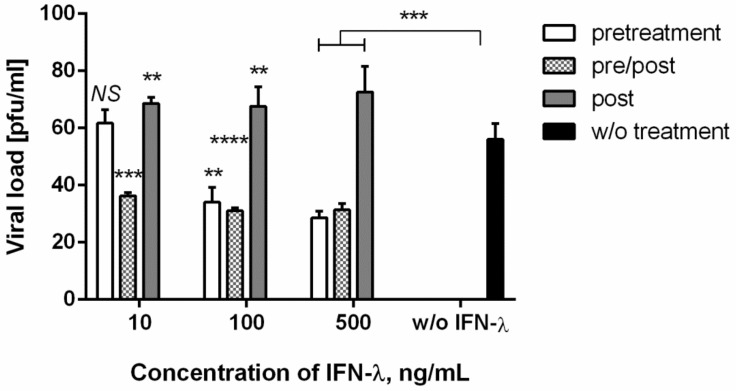
**Inhibition of CHIKV replication in different IFN-λ1 treatment regimes**. Vero cells were used. Virus was harvested at 24 hpi, and titers were determined by plaque assay. *p* value significance levels were determined for cells treated with each IFN-λ1 concentration, compared to untreated cells, by two-way ANOVA, followed by the pairwise Dunnett’s multiple comparisons test: ****―Adjusted *p* value < 0.0001; ***―<0.001; **―<0.01; NS—no significant differences. Data are represented as mean ± SD.

**Figure 5 viruses-13-01602-f005:**
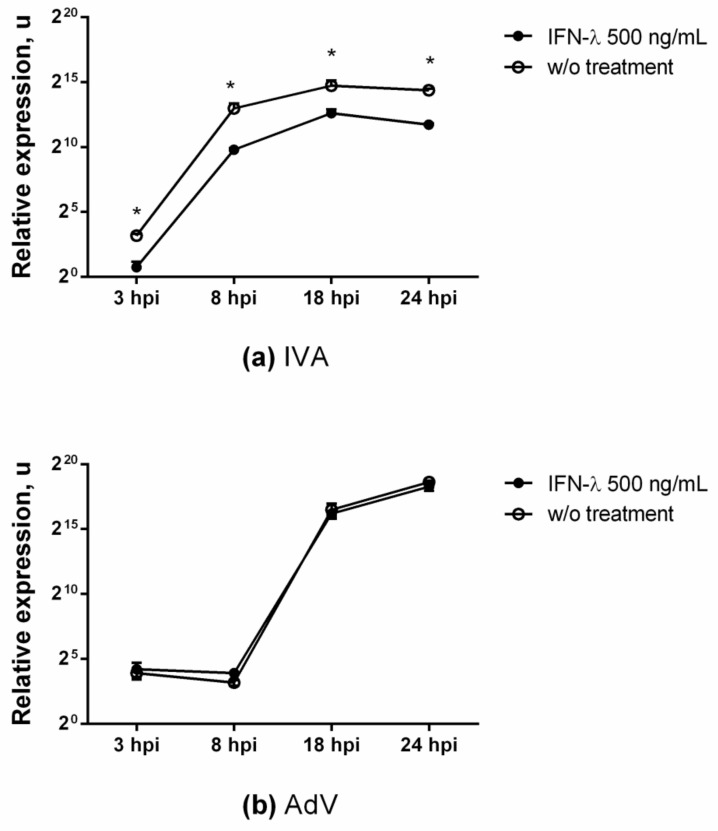
**Effect of IFN-λ1 treatment on replication of the IVA (**a**) and AdV (**b**) genomes**. Viral RNA yields (normalized to GAPDH) were determined after infection of A549 cells (untreated or IFN-λ1 treated at 500 ng/mL) with IVA or AdV (both 0.1 MOI). All data are representative of at least three independent experiments. *p* value significance levels were determined by unpaired nonparametric Mann–Whitney test (U-test). *—*p* value = 0.0286. Data are shown as mean ± SD.

## Data Availability

All data generated and analyzed during this study are included in this article.

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
