# Peer review of "IFN-λ1 Displays Various Levels of Antiviral Activity In Vitro in a Select Panel of RNA Viruses"

_viruses, 2021, doi:10.3390/v13081602_

Round 1

Reviewer 1 Report

General comment:

This study described the antiviral effects of one of the type III IFN, IFN-λ, suggesting multiple ISG expression induced by IFN-λ as their working mechanism. The authors have presented convincing evidence that IFN-λ treatment inhibits multiplication of several RNA viruses such as IAV, SARS-CoV-2, and CHIKV, but not a DNA virus.

The paper is well organized and clearly written. The results have been clearly presented. There are a couple of aspects that need to be tested to improve this work (see specific comments). 

Specific comments:

  1. Previously, it was reported that the IFN-λR1 chain of the type III IFN receptor complex can trigger apoptotic cell death, reducing cell growth. Thus, it is necessary to show that the range of concentration of IFN-λ tested in this study (10~500 ng/ml) has no cytotoxicity in both A549 and Vero cells.
  2. The authors show viral loads in the supernatant to describe antiviral effects of IFN-λ. It would be informative to include intracellular viral RNA levels in the same experimental conditions. As described in supplementary figure 4, IFN-λ did not induce the antiviral activity against adenovirus. RT-qPCR data might give an insight about a role of IFN-λ in viral RNA-dependent RNA polymerase activity regulation.

Reviewer 2 Report

IFN-λ is a more "recently-discovered" interferon. It has been studied in the antiviral field since their discovery in 2003. It has been shown before that  IFNλ mediates host antiviral response against different viruses including influenza viruses, coronaviruses, HBC, HCV, etc. This paper presented that IFN-λ exerted antiviral effect against influenza viruses and SARS-CoV-2 in vitro. In general, virus replication efficiency will reduce in the IFN-λ pre-treated cells.  The results are not surprising. The experiments are  straightforward and the presentation is overall clear.  However, it lacks novel conceptual advances for the field. It is a follow work of others and does not break new ground. 

Specific comments:

1. Line 132. Please briefly describe the method.

2. Figure 1A. NP ELISA only reflects the expression of NP protein. Standard titration assay is required to confirm the infectious virus replication.  e.g. plaque assay.

3. Figure 1A. Why the w/o treatment in post group is much lower than those in the other two groups?

3. Western blot and qPCR are required to show the viral protein and viral RNA or viral mRNA production.

4. Figure 1B/Figure 3 only showed one time point (72 h) which can be confounding because it is likely that the virus replications have reached plateau. The authors should conduct time-course experiment to show the virus growth kinetics in all the figures. 

5. Figure 2 shows a very small panel of ISGs. If the others cannot perform RNAseq, data from the public databases (e.g. GEO) should be found, analyzed and presented as supportive evidence.

6. Figure 3 shows the inhibition of SARS-CoV-2 replication. However, the therapeutic value is really low according to this experiment. 

7. In vivo study is required to demonstrate the antiviral functions of IFN-λ.

Round 2

Reviewer 1 Report

The revised manuscript has been improved and additional data support all previous comments.

Author Response

Thank you for your attention to our work.

Reviewer 2 Report

The authors have substantially addressed my concerns. There are a few points that can further improve this manuscript.

1) Figure 5 requires statistical analysis.

2) The authors need to add more introduction about human health thread by influenza viruses to strengthen their points including (a) The K526R substitution in viral protein PB2 enhances the effects of E627K on influenza virus replication. (b)An NS-segment exonic splicing enhancer regulates influenza A virus replication in mammalian cells. (c)The PB2 polymerase host adaptation substitutions prime Avian Indonesia Sub Clade 2.1 H5N1 Viruses for infecting humans. And etc.
